

# A feature selection method utilizing path accumulation cost, redundancy minimization, and interaction maximization for the diagnosis of coronary heart disease

Jiayao Jiang[1,*], Zheng Yue[1,*], Hongling Zhu[2], Yan Wang[2], Hongsen Cai[1] and Wenguang Hou[1]

[1] School of Life Science and Technology, Huazhong University of Science and Technology, Wuhan, China
[2] Division of Cardiology, Tongji Hospital, Huazhong University of Science and Technology, Wuhan, China
[*] These authors contributed equally to this work.

Corresponding authors
Hongsen Cai,
hongsencai@hust.edu.cn
Wenguang Hou,
houwenguang99@163.com

## ABSTRACT

**Background:** Coronary heart disease (CHD) is a major cause of mortality worldwide, with an increasing trend of affecting younger populations. The asymptomatic early stages and rapid progression of CHD make diagnosis challenging, necessitating efficient diagnostic approaches.

**Methods:** We propose a novel algorithm that focuses on accumulating soft path costs to discern crucial indicators from extensive diagnostic tests, aiming to improve early CHD identification. Our approach emphasizes feature interaction using an interaction accumulation evaluation function to identify features with maximal interaction and minimal redundancy. A new stopping criterion based on information gain ratio is also introduced.

**Results:** Experimental outcomes demonstrate that our algorithm outperforms several classical algorithms in terms of classification accuracy and feature dimension reduction, while also identifying highly correlated feature subsets.

**Conclusion:** The proposed approach offers an efficient solution for early detection of CHD by identifying critical indicators, reducing diagnostic complexity, and improving predictive accuracy, thus potentially leading to more effective CHD management.

# INTRODUCTION

Coronary heart disease (CHD) presents a significant global health challenge, accounting for approximately 17.9 million deaths annually. Myocardial infarctions and cerebrovascular accidents are primary contributors to CHD mortality, affecting individuals often without apparent symptoms in the early stages (*World Health Organization, 2024*). Increasing stress, unhealthy habits such as obesity, hypercholesterolemia, hypertension, and smoking have exacerbated the prevalence of CHD in younger populations (*Gonçalves et al., 2022*).

Early identification of CHD risk is crucial to prevent severe outcomes. However, conventional diagnostic methods, which rely on numerous overlapping markers, can be resource-intensive and inefficient (*Shahid et al., 2020*). Feature selection techniques offer promising approaches by reducing diagnostic markers, leading to improved screening models (*Li et al., 2020*).

## Related work

Recent advancements in machine learning have made it a powerful tool for the early diagnosis and prediction of CHD, particularly in handling high-dimensional medical datasets. Machine learning models, such as decision trees, random forests, and support vector machines, have been widely applied in feature selection for CHD prediction, as they are capable of capturing complex, non-linear relationships within the data.

Feature selection methodologies can be broadly categorized into filter, wrapper, and embedded approaches (*Zeng et al., 2015*). In early CHD prediction, wrapper and embedded methods have gained prominence. *Lin et al. (2022)* employed a combination of gradient boosting tree feature selection and recursive feature elimination to predict acute coronary syndrome risk, successfully extracting 25 key variables from 430 complex medical features, culminating in an impressive 98.8% accuracy in ACS risk prediction. Similarly, *Javeed et al. (2019)* integrated a random search algorithm (RSA) with an optimized random forest (RF) model, achieving a commendable 93.33% accuracy on the Cleveland datase. *Mohamed et al. (2020)* developed a novel Parasite-Predator Algorithm (PPA) by amalgamating the strengths of Cat Swarm Optimization (CSO), Cuckoo Search (CS), and Crow Search Algorithm (CSA). After PPA, fitness is used to select the feature subset that maximizes the accuracy and minimizes the number of selected features. This approach achieved an 86.17% accuracy on the Statlog dataset, ultimately selecting only four key features (*Mohamed et al., 2020*).

$$\text{Fitness} = \text{maximize}\left(\text{Acc} + w_f \cdot \left(1 - \frac{L_f}{L_t}\right)\right) \qquad (1)$$

where Acc is the classification accuracy, $L_f$ is the length of the selected feature subset, and $L_t$ is the total number of features.

Additionally, *Bharti et al. (2021)* conducted experiments on the UCI Heart dataset and found that combining feature selection with outlier detection yielded the best results. They applied LASSO for feature selection and trained models using machine learning and deep learning techniques. In the end, the deep learning model achieved an accuracy of 94.2%, while the KNN model achieved 84.8% (*Bharti et al., 2021*). *Fajri, Wiharto & Suryani (2023)* proposed a hybrid feature selection method combining Bee Swarm Optimization (BSO) and Q-learning (QBSO-FS), which improved heart disease detection accuracy, achieving 84.86% accuracy under the KNeighbors model, with faster convergence and better performance than traditional methods.

$$FC_s = F_s - \theta C_s \qquad (2)$$

where $FC_s$ is the evaluation value of feature selection, $F_s$ is the accuracy of feature selection, $C_s$ is the cost of feature subset, and $\theta$ is the adjustment coefficient.

Despite their targeted nature, wrapper and embedded methods have obvious limitations. Particularly, wrapper methods require extensive model training and evaluation in the search for an optimal feature subset, a process that becomes especially laborious in scenarios involving large datasets or an abundance of features. Conversely, embedded methods heavily rely on specific models, which may limit their generalizability.

In contrast, filter methods, which rapidly identify features related to the target variable through statistical theory and are computationally simple and independent of any machine classifier model, have emerged as a practical and efficient alternative for managing complex medical datasets, such as those used in early screening prediction for CHD. *Muhammad et al. (2020)* introduced a machine learning-based CHD detection model, employing four feature selection techniques with ten classification algorithms, significantly improving classification accuracy. *Reddy et al. (2023)* enhanced the performance of various classifiers in cardiac risk prediction by combining the Cleveland and Statlog heart datasets and employing principal component analysis (PCA) and correlation-based feature selection (CFS). *Peng, Long & Ding (2005)* proposed the mRMR method based on information theory, aimed at using mutual information between features to discern relevant and redundant features. *Yu & Liu (2003)* introduced a FCBF method, focusing on selecting main features and eliminating highly redundant features through the computation of SU and Markov blankets.

However, these filter methods primarily evaluate the association of individual features with classification, disregarding the synergistic impact of combined features on the prediction. This oversight could potentially lead to the erroneous exclusion of important features or the retention of irrelevant ones, particularly in complex applications like CHD risk prediction. To address this challenge, *Fleuret (2004)*, *Bennasar, Hicks & Setchi (2015)*, *Zeng et al. (2015)*, and *Wang, Jiang & Jiang (2021)* introduced various feature selection criteria (CMIM, JMIM, NJMIM, IWFS, MRMI), focusing on enhancing feature relevance, interaction, and redundancy removal, significantly improving the quality of feature selection and prediction accuracy. These advancements have collectively enhanced the quality of feature selection and prediction accuracy to a considerable extent. Notwithstanding the success of the above methods, they do not fully consider the relationships among all existing features and candidate features, potentially leading to the exclusion of relevant features or the inclusion of too many redundant features.

## Contribution

Existing early screening predictive models for coronary heart disease (CHD) often struggle with managing the intricate relationships between existing and candidate features. This issue may lead to the omission of important features or the inclusion of unnecessary redundancies. To address this issue, this article introduces an early screening predictive model based on a novel CHD feature selection algorithm. Our model emphasizes the interactions among multiple diagnostic markers and their linkage to CHD. Further, it selects a feature subset where the strong associations between features yield more valuable

information combinations, overcoming the above limitations and laying the groundwork for developing more sophisticated and effective CHD screening models. The main contributions are as follows:

1) We propose a feature selection approach based on soft path accumulative cost. This approach identifies the feature with the highest interaction by calculating the interaction of each current feature with all previous features. The final outcome is a subset of features with the strongest interrelationships, which best represent CHD.

2) To comprehensively evaluate the interactions among multiple features, we propose a novel interaction metric, Rel, and an accumulative interaction evaluation function SPA that measures the relationships between all existing features and the current candidate feature.

3) We introduce a novel stop criterion evaluation function to balance model evaluation performance and feature subset size. This function determines when to halt feature selection by measuring the increase in information gain and setting hyperparameters to balance the number of features with the final model evaluation performance.

The article is structured as follows: "Prior Knowledge of Feature Selection" introduces the background theoretical knowledge relevant to our discussion; "Methods" outlines the development process of our proposed early screening predictive model; "Results" describes the experimental setup and analyzes the findings. Finally, in "Discussion and Conclusion" we examines the implications of our study for improving CHD early screening and proposes directions for future advancements.

## PRIOR KNOWLEDGE OF FEATURE SELECTION

Before introducing the algorithm, we define the relevant formulas.

For a pair of discrete features $F_i, F_j$, the relationship between features is defined by Mutual Information (MI) $I(F_i, F_j)$:

$$I(F_i, F_j) = \sum_{F_i} \sum_{F_j} p(F_i, F_j) \log\left(\frac{p(F_i, F_j)}{p(F_i)p(F_j)}\right) \tag{3}$$

where $p(F_i, F_j)$, $p(F_i)$, and $p(F_j)$ are the probability density functions of $F_i, F_j$, $F_i$ and $F_j$, respectively.

Normalizing MI results in Symmetrical Uncertainty (SU), which is given by the following formula:

$$SU(F_i, F_j) = \frac{2 \cdot I(F_i, F_j)}{H(F_i) + H(F_j)} \tag{4}$$

where $H(F_i)$ and $H(F_j)$ are the entropies of $F_i$ and $F_j$, respectively.

Given that our proposed algorithm is based on feature interaction, we use the three-way interaction mutual information $I(F_i, F_j, C)$ to measure the relationship between a pair of discrete features $\{F_i, F_j\}$ and the label $C$ (*Tang, Dai & Xiang, 2019*).

$$I(F_i, F_j, C) = \sum \sum p(F_i, F_j, C) \log \left( \frac{p(F_i, F_j)p(F_i, C)p(F_j, C)}{p(F_i)p(F_j)p(C)p(F_i, F_j, C)} \right). \tag{5}$$

For $I(F_i, F_j, C)$, the selected feature $F_j$ should meet the condition of being relevant to the label $C$ and, in combination with the already selected feature $F_i$, providing more effective information (*Wang, Jiang & Jiang, 2021*), that is,

$$I(F_i, F_j, C) > I(F_i, C) + I(F_j, C) \quad \text{or} \quad I(F_i, F_j, C) > 0. \tag{6}$$

Markov Blanket is a criterion for judging the redundancy of a given feature $F_i$ in a feature set $F = \{F_1, F_2, F_3, \ldots, F_m\}$, defined as $M_i \in F(F_i \notin M_i)$. Mi is called the Markov Blanket of $F_i$ if and only if

$$P(F - M_i - \{F_i\}, C \mid F_i, M) = P(F - M_i - \{F_i\}, C \mid M_i). \tag{7}$$

If a given feature $F_i$ in the current feature set $F$ has its corresponding Markov Blanket $M_i$, it is considered redundant to the remaining features in the set.

For MI and SU, solely using mutual information $I(F_i, F_j)$ to assess the relationships between features can inadvertently bias the algorithm towards selecting features with higher entropy, which indicates more significant variability or complexity in their distribution. Although features with high entropy are information-rich, this does not inherently mean they share the closest relationship with the target variable. The normalization of MI through symmetrical uncertainty (SU) accounts for the entropies of both variables involved, thus ensuring that the metric no longer depends on the intrinsic information content of the variables themselves. By applying SU to normalize MI, the inherent bias associated with selecting features based on MI is rectified, shifting the focus towards the relative importance of features in relation to the target (*Yu & Liu, 2003*). Additionally, this normalization introduces an increased amount of feature information into the selection process.

On the other hand, unlike MI, the $I(F_i, F_j, C)$ does not need to be positive. If $F_i, F_j$ have a strong positive interaction, this means they can provide more information for the label $C$ than $F_i, F_j$ can each provide on their own. Moreover, for $F_i, F_j$, missing any one of the features would reduce the accuracy of the final classification result. If the interaction is negative or zero, it implies that $F_i$ and $F_j$ may offer redundant information or information irrelevant to the label $C$. Hence, an effective interaction between features should be where $I(F_i, F_j, C)$ is greater than zero and at its maximum.

## An information-theoretic perspective on feature selection

The fundamental goal of feature selection is to extract a key subset from the original feature set such that a classifier trained on this subset achieves classification performance comparable to, or even better than, that obtained using the full feature set. This goal can be formalized by the following conceptual function: under the constraint of classification performance, one seeks the smallest feature subset. That is,

$$\min_{S \subseteq \mathscr{F}} |S| \quad \text{subject to} \quad f(S) \geq f(\mathscr{F}) - \varepsilon, \tag{8}$$

where $f(S)$ denotes the accuracy of a classifier trained on the subset $S$ and $\varepsilon$ represents the permissible degradation in performance. This formulation not only encapsulates the requirement to maintain performance close to that of the full feature set but also underscores the objectives of dimensionality reduction and redundancy minimization.

Filter-based feature selection methods grounded in information theory assess two key aspects: the redundancy among features and the relevance of each feature to the target labels. Traditionally, mutual information (MI) is used for both purposes, with $I(F_i, F_j)$ measuring the redundancy between features $F_i$ and $F_j$, and $I(F_i, C)$ quantifying the relevance between a feature $F_i$ and the target $C$. However, MI's inherent limitations in scale and normalization have led to the adoption of symmetric uncertainty (SU) as a normalization tool, thereby enabling more consistent comparisons. Despite these improvements, the use of bivariate MI remains insufficient when trying to capture the additional joint gain achieved by combining a candidate feature with an already selected feature to explain the target. This deficiency motivates the introduction of three-way interaction mutual information, which is designed to account for the cooperative contribution of feature pairs in relation to the target.

The three-way interaction mutual information, denoted by $I(F_i, F_j, C)$, quantifies the additional information provided by the candidate feature $F_j$ in the presence of a selected feature $F_i$ when explaining the target $C$. Nevertheless, directly using this three-variable measure presents two issues. On one hand, it does not distinguish the redundancy that may exist between $F_i$ and $F_j$. On the other hand, the candidate feature $F_j$ might already exhibit high direct relevance to the target $C$, leading to potential double counting of its contribution.

To address these issues, we introduce two modulation terms based on symmetric uncertainty. The first modulation term is

$$\frac{I(F_i, F_j, C)}{SU(F_i, F_j)}, \tag{9}$$

which adjusts for the redundancy between $F_i$ and $F_j$. A high value of $SU(F_i, F_j)$ indicates substantial redundancy between the two features, thereby reducing the value of this term and reflecting that a large portion of the joint information is redundant.

The second modulation term is

$$\frac{I(F_i, F_j, C)}{SU(F_j, C)}, \tag{10}$$

which accounts for the direct relevance of the candidate feature $F_j$ to the target $C$. When $F_j$ is highly correlated with $C$, the high value of $SU(F_j, C)$ implies that much of the joint information is already captured by $F_j$'s individual relevance, and this contribution should not be overemphasized in assessing its unique gain.

Compared to conventional methods that rely solely on bivariate MI or SU to evaluate feature relevance and redundancy, this approach—employing three-way interaction mutual information modulated by SU offers a more comprehensive framework. It enables

a refined assessment of a candidate feature's joint gain, facilitating the selection of a compact feature subset that both preserves or enhances classification performance and reduces redundancy.

## METHODS

We proposed a feature selection algorithm based on feature correlation, redundancy, and interaction. We named this algorithm Soft Path Feature Selection (SPFS), which iteratively selects features with the highest interaction through path accumulation cost while minimizing redundancy as much as possible.

As illustrated in Fig. 1, the SPFS algorithm introduces an interaction metric, $\text{Rel}(F_i, F_j, C)$, which quantifies the interactions between features to balance their redundancy and relevance. Furthermore, the algorithm introduces an evaluation function called soft path accumulation (SPA). While ensuring the relevance of candidate features to the class label, it comprehensively considers relationships between all the selected features and each candidate feature, aiming to find the optimal balance between feature count and information.

The algorithm inputs a subset of candidate features and utilizes the Rel and the SPA to comprehensively assess each candidate feature's interaction with the previously selected best feature subset. It then selects the candidate with the strongest interaction to add to the current best subset. The feature selection process is iterative. At each step, the algorithm uses the selection stop scoring function score to determine whether to continue or stop. If a negative score appears, the algorithm records the current best feature subset and begins a count. If this count exceeds the predefined threshold, k_stale, the algorithm stops iteration and selects the subset with the highest score from the recorded best subsets as the final best feature subset.

### Quantitative evaluation of feature interactions and SPA evaluation function

From Eq. (4), it is evident that a newly added feature $F_j$, when combined with a previously selected best feature, should provide more information than each feature individually. However, as described in the section An Information-Theoretic Perspective on Feature Selection, highly correlated overlapping features might introduce excessive redundancy, which can be mitigated using symmetrical uncertainty (SU) to balance redundancy and relevance. Specifically, for a candidate feature $F_j$ and an already selected feature $F_i$, if $\text{SU}(F_j, C) < \text{SU}(F_i, F_j)$, this indicates that the relationship between $F_j$ and $F_i$ is stronger than that between $F_j$ and the label $C$. In such cases, $F_j$ and $F_i$ are considered to be highly redundant, indicating an overlap of information between $F_j$ and $F_i$.

Building on the characteristics of SU, we further propose an interaction metric, $\text{Rel}(F_i, F_j, C)$. $\text{Rel}(F_i, F_j, C)$ is defined as:

$$\text{Rel}(F_i, F_j, C) = \frac{I(F_i, F_j, C)}{SU(F_i, F_j)} - \frac{I(F_i, F_j, C)}{SU(F_j, C)}. \tag{11}$$

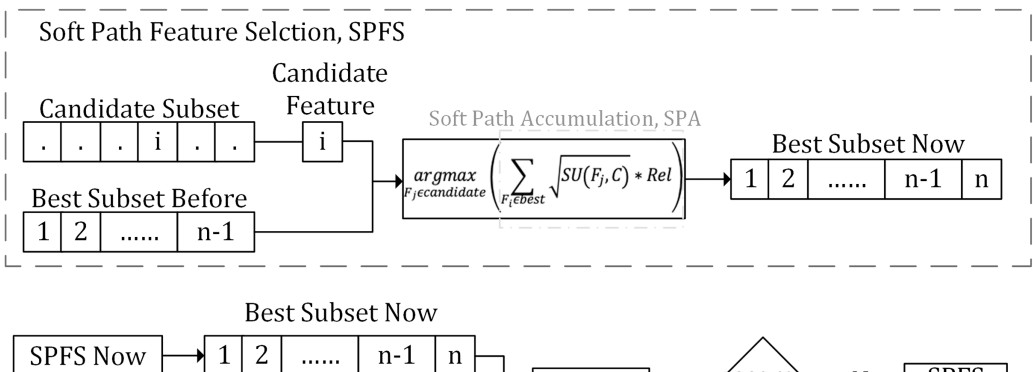

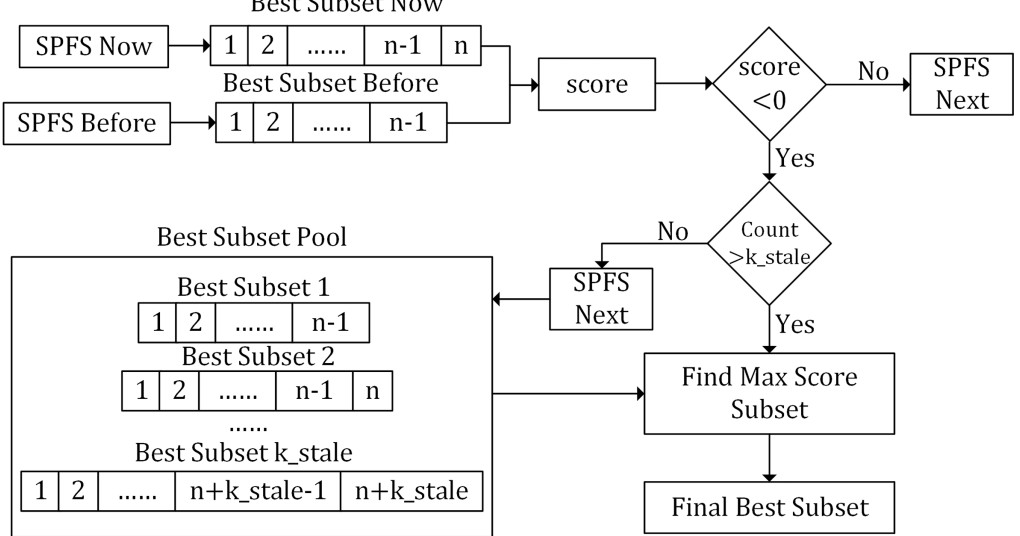

**Figure 1  Flowchart of the proposed method.** This flowchart illustrates the SPFS algorithm, which uses interaction metrics and soft path accumulation (SPA) to iteratively select features, balancing redundancy, relevance, and information to optimize the feature subset.

This metric evaluates the interactions between features by considering the three-way mutual information $I(F_i, F_j, C)$ and their symmetrical uncertainty SU, aiding in identifying feature combinations that may contribute limited predictive efficacy due to high redundancy.

Through $\text{Rel}(F_i, F_j, C)$, we can identify features $F_j$ that are highly related to $F_i$ yet exhibit low redundancy. However, focusing solely on $\text{Rel}(F_i, F_j, C)$ may lead to an overemphasis on the relationships among features without adequately considering the direct relevance of the candidate feature $F_j$ to the label $C$. Neglecting the direct association between candidate features and the label may result in a selected feature set that fails to establish effective predictive relationships, thereby impacting the model's predictive performance.

To address this issue, the SPA evaluation function is designed to consider both the interactions among features and their direct information contributions to the label. Specifically, the SPA scoring criterion incorporates the interaction measure $\text{Rel}(F_i, F_j, C)$ and explicitly includes the information contribution measure between the candidate feature $F_j$ and the label $C$. The SPA function evaluates the composite effect of a candidate feature on the entire best feature subset by cumulatively calculating the weighted

interactions between the candidate feature and all the features already selected in the best subset. The SPA scoring criterion is defined as:

$$\text{SPA} = \sum_{F_i \in \text{best}} \sqrt{\text{SU}(F_j, C)} \cdot \text{Rel}(F_i, F_j, C) \qquad (12)$$

where best is the subset of already selected best features, $F_j$ is the candidate feature, and $F_i$ is a previously selected best feature.

Based on the SPA evaluation function, we defined the SPFS selection strategy as follows:

$$\text{SPFS} = \arg \max_{F_j \in \text{candidate}} \left( \sum_{F_i \in \text{best}} \sqrt{\text{SU}(F_j, C)} \cdot \text{Rel}(F_i, F_j, C) \right) \qquad (13)$$

where candidate is the subset of candidate features. Through the SPFS scoring criterion, by considering the relevance of the current candidate feature $F_j$ to the label $C$ (weighted by $\text{SU}(F_j, C)$), we accumulate the interactive effects between the candidate feature and all previously selected features to select the candidate feature with the highest interaction with any feature in the best subset. This means that for the selected best features, the chosen candidate feature must demonstrate an unparalleled interaction with at least one feature in the best subset.

## Stop scoring function

Given the endless process of finding the highest interacting features, a reasonable stop scoring function is needed to identify an optimal feature subset that balances the number of features and prediction accuracy. We proposed that a feature should be included in the best subset if its addition significantly increased the information gain of the subset. The information gain ratio is extensively used to differentiate the contributions of features, because it reflects both the feature's entropy and the information gain contribution to the target variable, allowing us to precisely evaluate the effectiveness of different feature subsets.

The score is defined in Eq. (9), where $g_R(D, \text{best})$ and $g_R(D, \text{best\_before})$ are the information gain ratios of the current best feature subset and the previous best feature subset for the entire dataset $D$, respectively.

$$\begin{aligned} \text{score} &= g_R(D, \text{best}) - g_R(D, \text{best\_before}) \\ &= \frac{g(D, \text{best})}{H_{\text{best}}(D)} - \frac{g(D, \text{best\_before})}{H_{\text{best\_before}}(D)}. \end{aligned} \qquad (14)$$

The score is calculated to determine whether the current best feature subset provides a more significant information gain boost compared to the previous best feature subset. The conditions are as follows:

$$\text{score} \Rightarrow \begin{cases} \text{The newly selected feature brings useless information,} & \text{if score} < 0, \\ \text{The newly selected feature brings beneficial information,} & \text{if score} > 0. \end{cases} \qquad (15)$$

More precisely, Eq. (12) is explained in detail as follows:

- If score < 0, it indicates that the newly selected feature does not aid in the prediction accuracy and only increases the feature complexity.
- If score > 0, it suggests that the newly selected feature brings more beneficial information, aiding the model in making accurate predictions.

However, in some cases, the newly selected feature may not initially provide a significant boost in information gain, but as subsequent features are combined, it may produce better effects. To address this, we introduced a hyperparameter k_stale to delay the stop. When the number of times score < 0 reaches the threshold specified by k_stale, the selection of new features will be stopped. This approach allows for more flexible handling of early stopping issues caused by local minima in information gain.

## Procedures of soft path feature selection

As illustrated in Algorithm 1, given a dataset $D$ containing $n$ samples, $m$ features, and a class feature $C$, the SPFS algorithm identifies the best feature subset *best*. The SPFS feature selection process involves four primary stages: initialization, k_stale tuning, pre-computation, and feature selection.

In the initialization stage, the candidate feature subset, the best feature subset, and relevant algorithm parameters are initialized. During the pre-computation stage, the SU values for each feature and the mutual interaction Rel table between every pair of features and the label are calculated. The features are then ranked in descending order of their SU values to form the candidate feature subset.

The third stage focuses on tuning the hyperparameter $k\_stale$. Each $k\_stale\_value$ in $k\_range$ is applied during the feature selection phase, and the performance of each is evaluated to select the best $k\_stale$ for the final feature selection stage.

The fourth stage is a key step in the SPFS algorithm, and it's divided into two steps:

a. **Select individual features:** First, when the best feature subset is empty, the top-ranked feature from the sorted candidate feature subset is selected as the first feature of the best feature subset. In each iteration, the SPA evaluation function accumulates the interaction score of each candidate feature $F_i$ with the selected subset. Then, by using the SPFS selection strategy, the highest-scoring $F_{\text{best}}$ feature is chosen as the new feature in the best subset.

b. **Select the final best feature subset:** After each feature selection, the score is calculated; if the score is positive, iteration continues. When a negative score first appears, iteration continues, but it enters a delayed stopping step, counting the number of times the score is below zero and recording the best subset selected. When the number of times the score is below zero exceeds $k\_stale$, the addition of new features stops, and the algorithm selects the highest-scoring feature subset from these $k\_stale$ candidate best subsets as the final determined best subset.

Moreover, our algorithm also focuses on the stability of features. By observing whether the selected features remain consistent across multiple iterations and different data partitions, the stability of the feature selection process is assessed. A highly stable feature

**Algorithm 1  Feature selection algorithm.**

1: **Input:** Dataset $D$ with a full feature set $F$, the class $C$, $F = \{F_1, F_2, \cdots, F_m\}$, and the range of $k\_stale$
($k\_range$)

2: **Output:** Selected feature subset

3: **Initialization**

4: candidate $\leftarrow F$; best $\leftarrow \varnothing$; count $\leftarrow 0$; score $\leftarrow -\infty$; best_score $\leftarrow -\infty$;

5: $S_{SU} \leftarrow \varnothing$; $S_{Rel} \leftarrow \varnothing$; gain_least $\leftarrow []$;

6: **$k\_stale$ Tuning**

7: **for** each $k\_stale\_value$ in $k\_range$ **do**

8: $\qquad$ Perform the Feature-selection step using current $k\_stale\_value$;

9: $\qquad$ Update $best\_k\_stale$ if performance improves

10: **end for**

11: **Pre-computation**

12: **for** each $F_i, F_j \in F$ and $F_i \neq F_j$ **do**

13: $\qquad$ Calculate $S_{SU}[i] = SU(F_i, C)$;

14: $\qquad$ **if** $I(F_i, F_j, C) \geq 0$ **then**

15: $\qquad\qquad$ Set $S_{Rel}[i,j] = Rel(F_i, F_j, C)$

16: $\qquad$ **else**

17: $\qquad\qquad$ Set $S_{Rel}[i,j] = -\infty$

18: $\qquad$ **end if**

19: **end for**

20: Order candidate by $S_{SU}$ in descending order

21: **Feature-selection**

22: **while** candidate is not empty and count $< k\_stale$ **do**

23: $\qquad$ $F_i \leftarrow$ getFirstElement(candidate)

24: $\qquad$ Remove $F_i$ from candidate

25: $\qquad$ best $\leftarrow$ best $\cup \{F_i\}$

26: $\qquad$ **for** each $F_j \in$ candidate **do**

27: $\qquad\qquad$ Calculate SPA for $F_j$ considering current best

28: $\qquad\qquad$ Select the feature $F_{best}$ with the SPFS selection strategy, and best $\leftarrow$ best $\cup \{F_{best}\}$

29: $\qquad\qquad$ Remove $F_{best}$ from candidate

30: $\qquad\qquad$ Calculate score for current best

31: $\qquad\qquad$ **if** score $< 0$ **then**

32: $\qquad\qquad\qquad$ count $\leftarrow$ count + 1

33: $\qquad\qquad$ **end if**

34: $\qquad\qquad$ **if** count $> k\_stale$ **then**

*(Continued)*

| Algorithm 1 (continued) | |
|---|---|
| 35: | Select the best feature subset best with the largest score |
| 36: | **end if** |
| 37: | **end for** |
| 38: **end while** | |
| 39: End with the best feature subset best | |

selection process indicates that the selected features have a genuine and reliable association with the target variable.

Ultimately, the SPFS algorithm identifies an optimal feature subset *best* that is not overly complex but can effectively predict the class feature *C*, derived after considering feature importance, interaction, and feature stability. Through this approach, the algorithm not only enhances the prediction accuracy of the model but also avoids unnecessary model complexity, thereby providing an efficient and reliable feature selection solution for practical applications.

# RESULTS

In this section, we first design some experiments to evaluate the performance of the SPFS algorithm by making a contrast with other representative feature selection algorithms and then report the empirical results.

## Experiment setup

This research is dedicated to the development of an early screening and prediction model for coronary heart disease (CHD) that is particularly tailored for the young and middle-aged demographic, aiming to minimize the number of diagnostic markers needed. For this purpose, we analyzed four datasets associated with cardiac health and CHD. The first dataset, Heart, from the UCI Machine Learning Repository, features simplified binary classification of heart disease severity and excludes records with missing values. The Z-Alizadeh Sani dataset contains data from 303 patients at the Shaheed Rajaei Cardiovascular Center (*Alizadehsani et al., 2013*). Following this, Cardiovascular (*Lin et al., 2022*), from a tertiary hospital in Fujian, China, includes 2,702 patient records and focuses on predicting all-cause mortality, with all data anonymized. Lastly, CHD (*Cao et al., 2022*), approved by the ethics committee and anonymized, contains records from 715 patients at a tertiary hospital in Anhui, China. For both the Cardiovascular and Coronary Heart Disease datasets, they are open-source datasets from published articles that have already undergone ethical review in the original studies, and thus, this research does not require additional ethical approval.

Table 1 details the datasets used, including the number of features, cases, and categories. In the data preprocessing stage, since the obtained datasets had already undergone certain preprocessing steps by their original authors, we only implemented discretization to the datasets. Discretization aims to convert continuous numerical variables into discrete

**Table 1 Summary of benchmark datasets.**

| Dataset | Features | Samples | Positive | Negative | Classes |
|---|---|---|---|---|---|
| Heart | 13 | 297 | 137 | 160 | 2 |
| Z-Alizadeh Sani | 54 | 303 | 216 | 87 | 2 |
| Cardiovascular | 87 | 2,702 | 121 | 2,582 | 2 |
| Coronary heart disease | 43 | 725 | 262 | 453 | 2 |

categorical values (such as 0 and 1), facilitating subsequent feature selection and classification prediction. Specifically, we utilized the Minimum Description Length Principle (MDLP), a widely recognized discretization method proposed by *Fayyad & Irani (1993)*, to discretize all continuous numerical features. Additionally, for the imbalanced Cardiovascular dataset, the SMOTE technique was applied to address data imbalance issues during the classification stage, consistent with the methodology described in the original article. The implementation code for SPFS can be found at https://github.com/YUkiJiang559/SPFS.

To validate the efficacy of the proposed method, this study compared various widely used supervised feature selection algorithms, including FCBF (*Yu & Liu, 2003*), Consistency (*Dash & Liu, 2003*), mRMR (*Peng, Long & Ding, 2005*), CFS (*Hall, 2000*), Relief-F (*Urbanowicz et al., 2018*), CMIM (*Fleuret, 2004*), JMIM (*Bennasar, Hicks & Setchi, 2015*), IWFS (*Zeng et al., 2015*), and MRMI (*Wang, Jiang & Jiang, 2021*). These algorithms aim to identify the most relevant features, eliminate redundancies, and consider interactions among features to obtain a best feature subset. If the original articles of the datasets provided related feature selection results, these were also included for comparison. Among them, the CFS, FCBF, and mRMR algorithms obtain the best feature subset by finding the most relevant features and removing redundant features; the CMIM, JMIM, and IWFS algorithms focus more on the interaction between features to obtain the best feature subset, while MRMI uses the interaction between features while considering the relevance and redundancy of the features. Specifically, to ensure a fair comparison with the SPFS algorithm, CFS, Consistency, FCBF, and Relief-F were implemented using Weka's built-in library with default parameters (*Eibe Frank & Witten, 2016*). Similarly, CMIM, JMIM, and IWFS were implemented through the ITMO-FS library with default parameter settings (*Computer Technologies Laboratory, 2024*). It should be noted that the feature selection methods chosen for this study represent a set of well-established, conventional techniques that have been widely adopted in high-quality research publications as standard benchmarks for comparing feature selection performance.

## Empirical results

In line with the experimental design outlined earlier, we evaluated the performance of feature selection algorithms by examining the number of selected features and their classification accuracy. This evaluation involved classifying the best feature subsets with

XgBoost, naive Bayes, Linear-SVM, and random forest algorithms, all configured with default parameter settings (*Chen & Guestrin, 2016*; *Cervantes et al., 2020*; *Antoniadis, Lambert-Lacroix & Poggi, 2021*; *van de Schoot et al., 2021*). To ensure the stability and accuracy of the prediction results, ten-fold cross-validation was applied across all models. And two metrics were used to quantitative evaluation of feature selection algorithms: the algorithm's classification accuracy and its effectiveness in reducing feature dimensionality. Ideally, an algorithm that significantly surpasses others in classification accuracy is considered to have the best performance. However, when the difference in classification accuracy among algorithms is not significant, those that more effectively reduce feature dimensionality are deemed superior.

Among the involved algorithms, mRMR, Relief-F, CMIM, JMIM, and IWFS use a ranking procedure to select features, while others select the best feature subset based on their specific stopping criteria. Therefore, this study's experimental design aims to compare feature selection algorithms in two directions: firstly, for those algorithms determining the best feature subset based on their stopping criteria (CFS, Consistency, and FCBF), we compare their performance in reducing feature dimensions and the classification accuracy of their selected feature subset; secondly, for rank-based algorithms (mRMR, Relief-F, CMIM, JMIM, and IWFS), we evaluate the incremental gain in classification prediction accuracy for each feature selected in rank order. Considering the MRMI algorithm and the SPFS algorithm include both feature ranking and specific stopping criteria, this study evaluates their performance in both directions.

### Performance comparison with CFS, consistency, FCBF, and MRMI

Table 2 showcases the reduced number of features after applying the CFS, Consistency, FCBF, MRMI, and SPFS algorithms, with the columns for MRMI and SPFS specifically indicating the best feature subsets arranged in ranking order.

The experimental outcomes from Table 2 reveal that all feature selection algorithms achieved varying degrees of feature dimensionality reduction compared to the original datasets, with SPFS achieving the most significant reduction. This is particularly notable in the Cardiovascular dataset, where SPFS's reduction effect was the most pronounced. This suggests that SPFS can efficiently identify key features that improve the model's ability to distinguish between high-risk and low-risk patients, which is crucial in clinical decision-making for cardiovascular disease.

Subsequent tables (Tables 3, 4) detail the performance evaluation results of four classification models for the Heart, Z-Alizadeh Sani, Cardiovascular and Coronary Heart Disease, respectively. These results are presented as the average classification accuracy ± standard deviation from ten-fold cross-validation. The term "Full set features" in the first column denotes the performance results when all features of the original dataset are used for classification prediction, with the best feature selection algorithm results being highlighted in bold.

In this experimental section, Tables 3 and 4 highlight the effectiveness of the SPFS algorithm on four datasets. Similarly, in Heart and Z-Alizadeh Sani, SPFS consistently

**Table 2 Number of features selected with different algorithms.**

| Dataset | Full set features | CFS | Consistency | FCBF | MRMI | SPFS |
|---|---|---|---|---|---|---|
| Heart | 13 | 7 | 8 | 5 | 6 (13,9,12,8,3,10) | 3 (13,9,12) |
| Z-Alizadeh Sani | 54 | 11 | 11 | 10 | 5 (25,53,39,14,15) | 3 (25,53,1) |
| Cardiovascular | 87 | 19 | 18 | 9 | 5 (74,6,26,45,29) | 4 (74,6,18,71) |
| Coronary heart disease | 43 | 11 | 13 | 11 | 5 (41,1,30,28,29) | 5 (41,1,22,9,28) |

**Table 3 Classification accuracy (%) of heart and Z-Alizadeh Sani with selected features.**

| Algorithm | Dataset | Full set features | CFS | Consistency | FCBF | MRMI | SPFS |
|---|---|---|---|---|---|---|---|
| Xgboost | Heart | 75.20 ± 4.75 | 76.60 ± 4.65 | 74.20 ± 3.52 | 75.00 ± 4.58 | 76.20 ± 4.85 | **79.00 ± 3.71** |
| | Z-Alizadeh Sani | 81.37 ± 6.15 | 85.49 ± 4.98 | **88.04 ± 4.76** | 85.88 ± 4.37 | 75.88 ± 2.64 | 84.51 ± 4.84 |
| Naive Bayes | Heart | 74.40 ± 4.96 | 75.00 ± 5.88 | 74.60 ± 5.14 | 75.80 ± 5.02 | **76.00 ± 5.73** | 75.80 ± 5.02 |
| | Z-Alizadeh Sani | 41.18 ± 6.44 | 49.80 ± 6.46 | 45.88 ± 7.19 | 49.41 ± 6.84 | 42.35 ± 21.36 | **77.84 ± 2.91** |
| Linear SVM | Heart | 76.60 ± 4.74 | 76.80 ± 4.40 | 77.80 ± 4.24 | **80.40 ± 3.67** | 79.00 ± 4.02 | **80.40 ± 3.67** |
| | Z-Alizadeh Sani | 86.08 ± 5.07 | 83.92 ± 4.94 | 84.12 ± 4.15 | 84.31 ± 4.64 | 76.47 ± 2.77 | **87.06 ± 4.74** |
| Random Forest | Heart | 78.00 ± 4.47 | 76.00 ± 4.10 | 76.20 ± 3.94 | 73.80 ± 5.90 | 76.40 ± 4.36 | **79.20 ± 3.82** |
| | Z-Alizadeh Sani | 81.76 ± 4.48 | 83.92 ± 5.87 | **88.04 ± 4.15** | 85.29 ± 5.42 | 75.49 ± 3.19 | 84.51 ± 4.84 |
| Average | Heart | 76.05 ± 4.73 | 76.10 ± 4.76 | 75.70 ± 4.21 | 76.25 ± 4.79 | 76.90 ± 4.74 | **78.60 ± 4.05** |
| | Z-Alizadeh Sani | 72.60 ± 5.54 | 75.78 ± 5.56 | 76.52 ± 5.06 | 76.23 ± 5.32 | 67.55 ± 7.49 | **83.48 ± 4.33** |

**Note:**
The best results will be shown in bold.

**Table 4 Classification accuracy (%) of cardiovascular and coronary heart disease with selected features.**

| Algorithm | Dataset | Full set features | CFS | Consistency | FCBF | MRMI | SPFS |
|---|---|---|---|---|---|---|---|
| Xgboost | Cardiovascular | 95.68 ± 0.61 | 91.71 ± 0.96 | 91.84 ± 1.23 | 85.83 ± 1.86 | 86.90 ± 2.63 | **94.32 ± 0.64** |
| | Coronary heart disease | 95.75 ± 2.25 | 92.92 ± 1.46 | 93.25 ± 1.80 | 92.92 ± 1.46 | 81.67 ± 2.01 | **93.50 ± 2.52** |
| Naive Bayes | Cardiovascular | 93.75 ± 1.39 | 95.57 ± 0.47 | 95.45 ± 0.60 | 95.19 ± 0.79 | **96.85 ± 0.46** | 95.45 ± 0.91 |
| | Coronary heart disease | 84.58 ± 2.39 | **83.50 ± 2.13** | 82.25 ± 2.77 | **83.50 ± 2.13** | 81.25 ± 2.18 | 80.42 ± 2.02 |
| Linear SVM | Cardiovascular | 92.35 ± 0.41 | 88.38 ± 0.90 | 88.31 ± 1.47 | 79.67 ± 1.96 | 83.48 ± 3.16 | **94.28 ± 0.63** |
| | Coronary heart disease | 84.50 ± 2.56 | **83.17 ± 1.86** | 82.42 ± 1.99 | **83.17 ± 1.86** | 81.25 ± 2.18 | 80.42 ± 2.02 |
| Random Forest | Cardiovascular | 96.34 ± 0.44 | 90.84 ± 1.40 | 92.35 ± 0.90 | 85.43 ± 1.80 | 87.27 ± 2.71 | **94.32 ± 0.64** |
| | Coronary heart disease | 94.25 ± 2.12 | 92.83 ± 1.40 | 93.33 ± 2.36 | 92.92 ± 1.13 | 81.83 ± 2.13 | **93.42 ± 2.40** |
| Average | Cardiovascular | 94.53 ± 0.71 | 91.62 ± 0.93 | 91.99 ± 1.05 | 86.53 ± 1.60 | 88.63 ± 2.24 | **94.60 ± 0.71** |
| | Coronary heart disease | 89.77 ± 2.33 | 88.10 ± 1.71 | 87.81 ± 2.23 | **88.13 ± 1.64** | 81.50 ± 2.13 | 86.94 ± 2.24 |

**Note:**
The best results will be shown in bold.

recorded the highest average accuracy and, in most instances, achieved the highest classification accuracy. On the Heart dataset, SPFS achieved an average accuracy of 79.00%, surpassing CFS (76.60%), Consistency (74.20%), FCBF (75.00%), and MRMI (76.20%). Moreover, SPFS exhibited a remarkable capability for dimensionality reduction, achieving the highest reduction at 76.92%. This finding demonstrates that SPFS not only enhances accuracy but also helps streamline diagnostic processes, a significant advantage

in the clinical setting where fewer but more informative features can lead to faster decision-making. Remarkably, in the Z-Alizadeh Sani dataset, SPFS achieved significantly higher classification accuracy in the Naive Bayes evaluation than other methods, with $p$-values all less than 0.001. Analysis of the feature subsets selected by the feature selection algorithms revealed that CFS, Consistency, FCBF, and MRMI inevitably included features less relevant or not recommended by the dataset's original study (*e.g.*, features 14, 29, 28). Additionally, the performance of the feature subset from the original Z-Alizadeh Sani study in Xgboost, Naive_bayes, and Linear-svm models was inferior to SPFS, with a similar outcome in Random-forest. (Performance of the feature subset from the original study: Xgboost: 83.14%, Naive_bayes: 44.51%, Linear-svm: 83.92%, Random-forest: 85.10%) This further supports the notion that SPFS selects the most appropriate features that directly contribute to the accurate classification of patients, which is particularly beneficial in predicting coronary heart disease risk.

On the Cardiovascular dataset, SPFS outperformed the feature subsets selected in the original article across all evaluated models. Additionally, SPFS demonstrated $p$-values less than 0.001 for all models except naive Bayes. (Performance of the feature subset from the original study: Xgboost: 93.33%, Naive_bayes: 58.56%, Linear-svm: 87.83%, Random-forest: 93.37%) This highlights SPFS's capability in managing large, complex datasets while still retaining essential predictive information, making it a useful tool for clinical diagnosis.

Finally, for the Coronary Heart Disease dataset, the SPFS algorithm achieved a classification accuracy of 93.50% using the Xgboost classifier, highlighting its notable advantage in feature selection, with $p$-values all less than 0.01 compared to other methods. The SPFS showed exceptional performance in models handling complex data relationships, such as XGBoost and Random Forest. However, models which assume independence between features (naive Bayes) and are designed to handle linear relationships (Linear SVM) generated reduced performance, which means that the SPFS emphasized feature interactions and nonlinear relationships rather than selecting bad features. Unlike other algorithms, which did not account for interactions between multiple features, SPFS's consideration of feature interactions contributed to its superior performance. While SPFS may not have achieved the highest average accuracy, its ability to effectively reduce dimensionality and select the most relevant features underscores the critical role of feature selection strategies in enhancing model performance. The original article utilized 24 features, whereas SPFS opted for only 5, showcasing SPFS's excellence in reducing feature count while preserving high performance. (Performance of the feature subset from the original study: Xgboost: 94.58%, Naive_bayes: 83.17%, Linear-svm: 83.83%, Random-forest: 93.50%) This demonstrates that SPFS can identify the core features crucial for diagnosing coronary heart disease while eliminating unnecessary ones, which is crucial for improving clinical efficiency. Furthermore, when compared to using all features in model predictions, SPFS consistently demonstrated superior accuracy across all datasets, except for the Coronary Heart Disease dataset.

**Table 5 Top-10 ranked features by different algorithms.**

| Methods | Heart | Z-Alizadeh Sani | Cardiovascular | Coronary heart disease |
|---|---|---|---|---|
| mRMR | 13, 12, 3, 11, 9, 2, 8, 10, 1, 4 | 25, 54, 1, 35, 53, 6, 28, 18, 24, 32 | 75, 49, 69, 29, 45, 72, 26, 81, 56, 74 | 9, 41, 22, 28, 16, 43, 29, 1, 23, 33 |
| Relief-F | 13, 2, 9, 12, 8, 3, 11, 10, 6, 1 | 25, 28, 7, 53, 26, 8, 1, 35, 4, 6 | 10, 56, 57, 40, 5, 4, 6, 8, 3, 2 | 22, 41, 1, 12, 28, 43, 16, 18, 38, 11 |
| CMIM | 13, 12, 3, 9, 11, 2, 10, 8, 7, 6 | 25, 54, 35, 53, 18, 6, 1, 55, 32, 38 | 75, 49, 45, 20, 29, 69, 26, 57, 44, 21 | 9, 41, 22, 28, 17, 16, 1, 33, 27, 18 |
| JMIM | 13, 12, 3, 9, 11, 2, 10, 8, 1, 7 | 25, 54, 28, 7, 53, 55, 35, 1, 29, 6 | 75, 49, 69, 45, 81, 71, 76, 70, 20, 29 | 9, 41, 22, 28, 43, 17, 23, 27, 16, 29 |
| IWFS | 13, 12, 3, 9, 10, 11, 2, 8, 1, 7 | 25, 54, 28, 53, 24, 35, 1, 29, 6, 32 | 74, 81, 32, 72, 75, 69, 80, 38, 22, 77 | 41, 22, 28, 9, 43, 29, 23, 33, 17, 16 |
| MRMI | 13, 9, 12, 8, 3, 10 | 25, 53, 39, 14, 15 | 74, 6, 26, 45, 29 | 41, 1, 30, 28, 29 |
| SPFS | 13, 9, 12 | 25, 53, 1 | 74, 6, 18, 71 | 41, 1, 22, 9, 28 |

***Performance comparison of mRMR, Relief-F, CMIM, JMIM, IWFS, and MRMI***
For algorithms that select features based on a ranking procedure, by specifying a required number of features $N$, the feature selection algorithm outputs the top $N$ best features to form the final most important feature ranking subset. For the four datasets covered in the experiment, the required number of features $N$ was set to 10. For these datasets, each algorithm ranked the features by importance and provided the corresponding most important feature ranking subset. Table 5 lists the different algorithms selected features in order of importance. We can observe that although the ranking orders vary, certain key features consistently appear at the top across the subsets of different algorithms. These commonly deemed important features form the core of the SPFS algorithm's selection. This indicates that while each feature selection algorithm can capture the most important information among many features, it may also capture incorrect information or noise. Meanwhile, the SPFS algorithm exhibits significant efficiency in feature selection, adeptly identifying the most crucial features within complex information while disregarding irrelevant ones. This underscores the SPFS algorithm's effectiveness in capturing essential data within datasets.

The experimental results displayed in Fig. 2 indicate that the accuracy of almost all feature selection algorithms improves with an increase in the number of features. However, after a certain point, the performance improvement of the algorithms plateaus or even declines, highlighting the importance of selecting an optimal number of feature subsets to enhance model performance.

Comparing with algorithms that select features based on a ranking procedure, it can be seen that the results obtained using the SPFS method are very close or similar to those obtained with mRMR, Relief-F, CMIM, JMIM, IWFS, and MRMI methods, and sometimes even better.

In the Heart and Cardiovascular datasets, SPFS was almost the best-performing algorithm. Besides SPFS, other algorithms exhibited significant performance drops or sharp declines after initial improvements. The sudden decline and subsequent improvement in the performance of the SPFS algorithm on the Heart dataset may be because the second feature alone did not contribute much information. However, it has a strong mutual correlation with the third feature, leading to a significant increase in

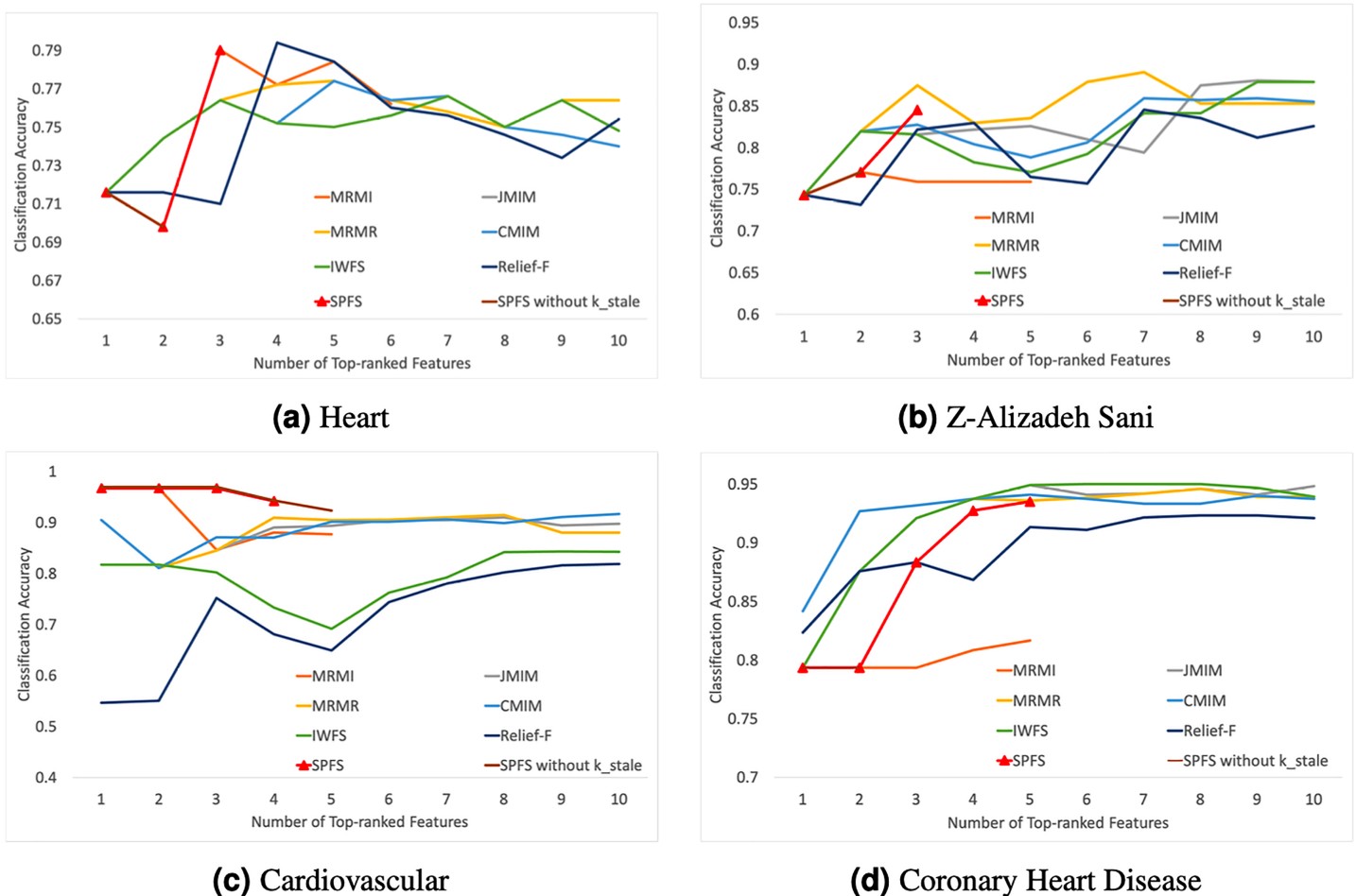

**Figure 2 Average classification accuracy vs. number of top-ranked features.** (A)–(D) represent different data sets, and the label is followed by the name of the corresponding data set. Figure shows that SPFS achieves competitive or superior performance with fewer features, while other algorithms' accuracy plateaus or declines after a certain point.

accuracy performance once the third feature was included. This can be attributed to the effectiveness of the $k\_stale$ parameter, which prevents premature stopping, allowing the algorithm to continue selecting features that provide additional useful information when combined with previous ones. As can be seen by the SPFS without $k\_stale$ in the line graph, Heart will stop at the second feature without $k\_stale$. In the Z-Alizadeh Sani dataset, SPFS's steady improvement was second only to the MRMR algorithm, while other algorithms required more features to achieve comparable or higher accuracy levels, contrary to our objective of reducing feature dimensions. In the Coronary Heart Disease dataset, the SPFS algorithm's performance steadily increased with the addition of features. Although the final result was not the best, it was similar to the performances of other superior algorithms, which did not show much improvement or even deteriorated with additional features. In the Cardiovascular dataset, the $k\_stale$ parameter effectively stopped the feature selection before accuracy declined sharply, ensuring that the selected feature subset remained highly effective.

In summary, the SPFS algorithm can select the most important and accurate features with a lower required number of features, achieving rapid improvements in accuracy while balancing high accuracy and low feature dimensionality. Comparisons with the other six feature selection methods show that SPFS can achieve better or similar performance with fewer features, proving its superiority.

## DISCUSSION AND CONCLUSION

In our study, we introduced SPFS, a pioneering approach to feature selection that intricately assesses the relevance, redundancy, and interactivity among features. By computing the path cumulative cost, SPFS adeptly minimizes redundancy while pinpointing the most interactively significant features, demonstrating exceptional efficacy across diverse datasets. One notable finding emerged when applying SPFS to large-scale, highly imbalanced cardiac datasets, where the algorithm not only maintained classification accuracy but also enhanced it by selecting a smaller subset of features. This success is likely due to SPFS's unique ability to account for the complex interactions between features, ensuring that even in the presence of extreme class imbalances, the selected features retain predictive power. This finding highlights a gap in current medical data analytics, where methods for handling imbalanced datasets are often insufficient or underdeveloped. Despite the significant prevalence of class imbalances in medical datasets, many existing techniques are not well-suited to address this challenge. SPFS, however, demonstrates its potential by successfully handling such imbalances.

However, SPFS is not without its limitations, notably its dependence on precise parameterization. Its performance is markedly influenced by the critical parameter, $k\_stale$, underscoring the importance of accurate parameter configuration for achieving the best feature subset selection. The parameter tuning process, often reliant on exhaustive search techniques, demands an in-depth understanding of the dataset, which can be daunting for users with limited domain expertise. Additionally, the algorithm's sensitivity to parameter changes might hinder its broad applicability.

Looking ahead, we aim to expand the use of SPFS to a wider range of medical health datasets, especially those with extreme class imbalances. Furthermore, we plan to improve SPFS's handling of such imbalances by explicitly incorporating information about the differences between positive and negative class distributions in the training data. This could involve developing strategies that adaptively select features based on these class disparities, further improving its performance in large-scale, imbalanced datasets. Additionally, we will integrate SPFS with cutting-edge machine learning techniques to enhance its potential for high-dimensional and complex data structure recognition and feature extraction. Future developments will focus on improving algorithm adaptability by automating parameter adjustments, potentially using reinforcement learning or meta-learning approaches. Structural improvements will also be made to more effectively process large-scale, complex datasets, allowing SPFS to better handle the diverse challenges posed by real-world medical data.

### Funding

This work was supported by the National Key Research and Development Program (No. 2022YFE0209900) and the Hubei Provincial Key Research and Development Program (2023BCB007). The funders had no role in study design, data collection and analysis, decision to publish, or preparation of the manuscript.

### Grant Disclosures

The following grant information was disclosed by the authors:
National Key Research and Development Program: 2022YFE0209900.
Hubei Provincial Key Research and Development Program: 2023BCB007.

### Competing Interests

The authors declare that they have no competing interests.

### Author Contributions

- Jiayao Jiang conceived and designed the experiments, performed the experiments, analyzed the data, performed the computation work, prepared figures and/or tables, authored or reviewed drafts of the article, and approved the final draft.
- Zheng Yue conceived and designed the experiments, performed the computation work, authored or reviewed drafts of the article, and approved the final draft.
- Hongling Zhu conceived and designed the experiments, prepared figures and/or tables, and approved the final draft.
- Yan Wang conceived and designed the experiments, prepared figures and/or tables, and approved the final draft.
- Hongsen Cai analyzed the data, prepared figures and/or tables, and approved the final draft.
- Wenguang Hou conceived and designed the experiments, authored or reviewed drafts of the article, and approved the final draft.

### Data Availability

The data and code is available at GitHub: https://github.com/YUkiJiang559/SPFS.git.

The Heart Disease dataset is available at UCI: Janosi, A., Steinbrunn, W., Pfisterer, M., & Detrano, R. (1989). Heart Disease [Dataset]. UCI Machine Learning Repository. https://doi.org/10.24432/C52P4X.

The Z-Alizadeh Sani dataset is available at UCI: Alizadehsani, R., Roshanzamir, M., & Sani, Z. (2013). Z-Alizadeh Sani [Dataset]. UCI Machine Learning Repository. https://doi.org/10.24432/C5Q31T.

The Cerebrovascular data is available at GitHub: https://github.com/xueyutao/Cerebrovascular.

The Coronary Heart Disease data is available at PeerJ: 10.7717/peerj.14078/supp-1.

## Supplemental Information

Supplemental information for this article can be found online at http://dx.doi.org/10.7717/peerj-cs.2834#supplemental-information.

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
