# Peer review of "A feature selection method utilizing path accumulation cost, redundancy minimization, and interaction maximization for the diagnosis of coronary heart disease"

_PeerJ Computer Science, doi:10.7717/peerj-cs.2834_

## Round 0.1 · original submission · Major Revisions

The authors should carefully read and consider the reviewers' comments. In particular, Reviewer 1 has commented on the importance of highlighting the broader impact of SPFS and explaining the advantages of verifying SPFS across various datasets or applications to support replication. Reviewer 3 has clearly provided an enumerated list of suggested modifications that the authors should implement in the manuscript.

·

Basic reporting

Detailed Review on Basic Reporting

Technical concepts like "SPFS," "Symmetrical Uncertainty (SU)," and "Interaction Maximization (IM)" are defined consistently throughout the article, which is written clearly and professionally. Because most sections are brief and straightforward, even an academic audience may understand the content. However, certain passages with complex phrases could make it difficult for readers outside of the immediate subject to understand, especially in the Introduction and Methods sections. For instance, separating the specifics of the formula from its consequences would simplify the explanation of "Symmetrical Uncertainty" (lines 120–128). Similarly, the stop-scoring function description (line 200 onward) is accurate but might be made more explicit by segmenting it into smaller parts. The clarity for readers would be increased by contextualizing equations in simpler words and providing algorithmic explanations in bullet points or numbered lists to increase accessibility.
The paper comprehensively analyzes previous feature selection techniques, including embedded techniques like MRMI and conventional techniques like mRMR and Relief-F. It successfully points out shortcomings in earlier research, like the scant attention given to feature interactions in intricate datasets, which supports the creation of the SPFS algorithm. The Related Work section, however, might better highlight how current algorithms, especially those that diagnose CHD or other comparable applications, function. Although the limitations in feature selection are clearly stated, there is room for a more explicit discussion of the relationship to CHD datasets. This section would benefit from a table summarizing the main feature selection techniques linked to CHD, their drawbacks, and how SPFS handles them.
The Introduction, Methods, Results, and Discussion sections are clearly defined, and the work follows professional structural guidelines. Tables and figures are understandable, pertinent, and educational. While Figure 1 offers a helpful flowchart of the SPFS method, Tables 3 and 4 provide a practical summary of classification accuracy comparisons. More thorough captions that describe the trends, including the plateau effect and performance peaks, would be helpful for Figure 2, which plots categorization accuracy vs the amount of features. Although tables do a good job summarizing performance data, comparisons would be more enlightening if they included a column or row highlighting statistical significance, if relevant. Interpretability would also be enhanced by including succinct explanations in figure captions.
The self-contained manuscript includes enough background information, methodology, and findings to support research replication. The findings are closely related to the theory. Although enhancing early CHD detection is obvious, it might be mentioned explicitly in the Results section to relate the findings to the research goals better. Furthermore, there is no visual representation of the feature selection outcomes. Including a chart or table that summarizes its impact across datasets would be insightful.
The article offers precise definitions and mathematical calculations for ideas like Mutual Information (MI), Symmetrical Uncertainty (SU), and the SPA evaluation function. Some explanations, such as the stop-scoring function, are conceptual and lack solid mathematical support. Although not explicitly shown, the connection between feature interaction metrics and prediction accuracy is mentioned. The theoretical underpinnings of the study would be strengthened by providing formal proofs or derivations for essential ideas like the stop-scoring function and SPA evaluation. These facts could be included in an appendix if space is limited.

Summary of Strengths and Recommendations

Clarity, a comprehensive literature analysis, a solid contextualization of SPFS's originality, and adherence to professional structural norms are all excellent aspects of the article. It contains pertinent, excellent figures and tables and gives findings that agree with the hypotheses. Simplifying complex portions, adding comparisons unique to CHD, and adding more explicit mathematical proofs are ways to improve the Related Work section. Enhancing figure captions and visualizing the impact of k-stale would help improve the presentation. These improvements will increase the manuscript's impact and readability for many scholars.

Experimental design

Review of Experimental Design

The article presents original primary research that creates a novel feature selection method specifically designed for early coronary heart disease (CHD) diagnosis, which aligns with the journal's goals and scope. The study tackles a significant computational barrier in analyzing intricate medical information by concentrating on feature selection that minimizes redundancy and maximizes interactions. This creative method advances medical data analytics and computational science. Although the publication does a good job illustrating its applicability to computational approaches, it might do a better job of stressing the usefulness of SPFS for clinical decision-making and enhancing diagnostic results for medical researchers and clinicians.

The study aims to create a feature selection technique that balances between minimizing redundancies and maximizing interactions, specifically to enhance the diagnosis of congenital heart disease. The paper points out essential flaws in current approaches, such as their inability to adequately eliminate redundancy or account for feature interactions, particularly in datasets with complex relationships. The suggested SPFS method closes a known knowledge gap by resolving these drawbacks. The research's wider ramifications aren't highlighted enough. The effect of the manuscript would be increased by including such a debate.

To validate SPFS, the experimental design is rigorous and uses a variety of datasets with different feature dimensions and sample sizes. A thorough examination is ensured by including ethically approved and publicly accessible medical datasets. According to the manuscript, ethics permissions were acquired, and data were anonymized, demonstrating respect for ethical norms. It does not, however, detail the precise ethical issues considered, such as data processing procedures or adherence to laws like GDPR or HIPAA. The study's ethical compliance would be more transparent if these factors were briefly described.

Enough information is provided about the procedures to enable replication. Preprocessing stages like the Minimum Description Length Principle for discretization are explained in detail, as are essential components of the SPFS algorithm like the SPA evaluation function and the halting condition. The study also compares using several benchmark methods (such as mRMR, Relief-F, and MRMI). Nevertheless, some implementation specifics, including particular hyperparameter ranges and the software tools or libraries used, are not explicitly explained. Additionally, complete replicability could be restricted by the absence of code or additional elements. This problem could be resolved, and transparency could be increased by publishing code or providing a link to the GitHub repository.

Recommendations

To strengthen the manuscript, the authors could specifically address the broader clinical implications of SPFS, such as how it might assist risk stratification in CHD or diagnostic procedures. More information about ethical procedures, like data anonymization techniques and adherence to pertinent laws, should also be included. Mentioning the software tools or libraries used and providing a more thorough explanation of the hyperparameter tuning procedures would further support replication. Sharing code or additional resources would also substantially improve transparency and reproducibility.

In conclusion, the study is carefully planned, meticulously carried out, and pertinent to the journal's focus. The well-defined research question addresses a significant gap in feature selection for the diagnosis of congenital heart disease. Implementing the suggested improvements would strengthen the manuscript's overall effect, replicability, and rigor.

Validity of the findings

Review of the Validity of the Findings

The study's conclusions illustrate the efficacy of the SPFS (Soft Path Feature Selection) method in resolving issues related to feature redundancy and interaction in high-dimensional datasets, exhibiting both novelty and impact. The algorithm's capacity to decrease dimensionality while preserving or enhancing classification accuracy is highlighted in the publication, which represents a substantial improvement over current techniques like mRMR and Relief-F. It does not, however, specifically address SPFS's broader effects on the industry, such as its potential to establish new standards for feature selection in the medical or other fields. Furthermore, even though SPFS's innovation is implied in the text, a more direct comparison would show how the algorithm addresses particular drawbacks of earlier methods.
The study's justification is presented, emphasizing the necessity of a compelling feature selection technique for early coronary heart disease (CHD) detection. The experimental design guarantees meaningful replication by offering thorough explanations of the methods, including the computation of interaction metrics and the SPA evaluation function. These specifics enable others to repeat the study's methodology. Nevertheless, the advantages of replication for the larger body of literature—like confirming SPFS for other datasets or domains—are unclear. Although the technique is thorough, the ease with which other researchers can replicate the findings is limited by the lack of implementation code or tools.
The study's data are reliable because they were taken from several datasets with different samples and feature sizes. This diversity supports the findings' generalizability. Ten-fold cross-validation and comparisons between several classifiers, including Random Forest and XGBoost, are also used to guarantee statistical soundness. These methods successfully reduce the possibility of biased findings from arbitrary data splits. However, the study does not cover potential problems brought about by preprocessing stages, including discretization or the methods used to correct for outliers and data imbalances. The absence of these elements is a serious flaw because they can significantly impact outcomes, particularly in medical datasets.
The study's conclusions are clearly expressed, pertinent to the research issue, and suitably restricted to the data shown. The fact that SPFS reduces dimensionality without compromising classification performance is skillfully highlighted in the study. Nonetheless, a more explicit link between the study's findings and their implications for early CHD detection may be made in the conclusions. For instance, discussing how SPFS could enhance clinical decision-making or diagnostic procedures would be helpful. Additionally, to underscore SPFS's potential influence beyond CHD diagnosis, its broader applicability to other sectors might be highlighted.

Recommendations

To bolster the validity of the findings, the authors should specifically explore the broader impact of SPFS, especially how it fills in noted holes in current approaches and its implications for other areas. They should also clearly explain the advantages of verifying SPFS across various datasets or applications to support replication. Discussing the management of preprocessing procedures, outliers, and imbalances to guarantee objective outcomes is necessary to address statistical soundness in more detail. Lastly, the study's significance and transparency would be increased by connecting the findings to real-world applications for diagnosing congenital heart disease and making the algorithm's implementation accessible.
To sum up, the results are solid, statistically sound, and consistent with the study's goals. The manuscript's argument for the importance and suitability of the SPFS algorithm in medical and other computational contexts would be strengthened if the aforementioned suggestions were considered.

Reviewer 2 ·

Basic reporting

Clear and unambiguous. Correct English grammar used throughout the entire manuscript.

For the introduction, I suggest that the authors first introduce or do mention the role of machine learning models as screening methods for CHD before focusing on the use and/or benefits of feature selection algorithms.

The literature review has been simplified but rather extensive as the authors emphasized the pros and cons of existing feature selection algorithms.

Experimental design

There is exhaustive discussion on the proposed feature algorithm (SPFS) including its underlying theoretical and mathematical assumptions. Likewise, the needed metrics to evaluate this SPFS is clearly indicated (via Rel, SPA, and stop scoring function). This clearly indicates that there is rigorous investigation or technical assessment of the proposed algorithms.

Additionally, the use several datasets (publicly available and to test the SPFS is commendable particularly the two datasets from China (which are not publicly available) but nonetheless, ethical approval have been obtained.

However, some datasets have moderate to severe imbalance and the authors did not specify whether the imbalance have been addressed. Also, authors did not clearly indicate whether hyperparameter tuning was performed on all models across several datasets.

The authors also analyzed several feature selection algorithms to compare with SPFS (mRMR, Relief-F, CMIM, JMIM, IWFS, etc.). I agree with their decision to evaluate the algorithms that determine best feature subset based on the stopping criteria via performance in reducing feature dimension and classification accuracy of the selected feature subset as well as for those rank-based algorithms via incremental gain in classification accuracy for each feature.

Validity of the findings

In the presentation of results, particularly as shown in Table 3, the authors did specify that the proposed SPFS did achieve higher classification accuracy against the other methods (full features, CFS etc) across several machine learning models. However, I suggest to avoid the use of significantly higher metrics is statistical test to assess significance was not employed.

The textual description of the results in Table 4 seem to be off. The values do not match those seen in Table 4. There is a need to recheck the comparison made for Table 4.

Discussion of the results is satisfactory as the analysis focused on the supporting evidence as to classification accuracy of the SPFS. However, the limitations of the study are not specified more particularly, the heart datasets with data imbalance. There is a need to highlight this limitation of the study and suggest future research steps to address these limitations.

Additional comments

Suggest to accept the manuscript with minor revisions.

·

Basic reporting

1. The main problem is that it appears that the paper does not start with a hypothesis to test, or research question to answer but rather focused solely on applying a machine learning method to the data. Author can write research question to answer
2. Authors need to add details on inclusion and exclusion criteria for participants
3. Authors need to explain attribute and dataset
4. Authors need to explain data pre-processing tasks and result
5. Some of the existing similar studies have reported higher performance compared to this study; therefore, it is necessary to include a discussion on the difference.
6. Authors need to write limitation of this research

Experimental design

1. The main problem is that it appears that the paper does not start with a hypothesis to test, or research question to answer but rather focused solely on applying a machine learning method to the data. Author can write research question to answer
2. Authors need to add details on inclusion and exclusion criteria for participants
3. Authors need to explain attribute and dataset
4. Authors need to explain data pre-processing tasks and result
5. Some of the existing similar studies have reported higher performance compared to this study, therefore it is necessary to include a discussion on the difference.
6. Authors need to write limitation of this research

Validity of the findings

1. The main problem is that it appears that the paper does not start with a hypothesis to test, or research question to answer but rather focused solely on applying a machine learning method to the data. Author can write research question to answer
2. Authors need to add details on inclusion and exclusion criteria for participants
3. Authors need to explain attribute and dataset
4. Authors need to explain data pre-processing tasks and result
5. Some of the existing similar studies have reported higher performance compared to this study, therefore it is necessary to include a discussion on the difference.
6. Authors need to write limitation of this research

---

## Round 0.2 · Minor Revisions

Please address the comments made by Reviewer #3. Specifically, a hypothesis or research question should be included. A brief comparison with existing studies that have reported higher/better performance should be addressed.

Please included a detailed summary of any changes you are making.

·

Basic reporting

The manuscript was accurately revised.

Experimental design

no comment

Validity of the findings

no comment

·

Basic reporting

1. The main problem is that it appears that the paper does not start with a hypothesis to test, or research question to answer but rather focused solely on applying a machine learning method to the data. Author can write research question to answer
2. Authors need to explain data pre-processing tasks and result
3. Some of the existing similar studies have reported higher performance compared to this study, therefore it is necessary to include a discussion on the difference.
4. Authors need to write limitation of this research

Experimental design

2. Authors need to explain data pre-processing tasks and result

Validity of the findings

3. Some of the existing similar studies have reported higher performance compared to this study, therefore it is necessary to include a discussion on the difference.
4. Authors need to write limitation of this research

Additional comments

1. The main problem is that it appears that the paper does not start with a hypothesis to test, or research question to answer but rather focused solely on applying a machine learning method to the data. Author can write research question to answer
2. Authors need to explain data pre-processing tasks and result
3. Some of the existing similar studies have reported higher performance compared to this study, therefore it is necessary to include a discussion on the difference.
4. Authors need to write limitation of this research

---

## Round 0.3 · accepted · Accept

The authors have adequately addressed the reviewers' comments. Version 1 of the manuscript has been re-reviewed by two of the original reviewers, and the authors have addressed these reviewers' comments as well.

I believe that this manuscript is ready for publication.